# Proposal for Applying Sustainable Drainage Systems (SuDSs) as a Strategic Business Unit at a Military Development Located in Southern Europe (Córdoba, Spain): "Project BLET"

**Antonio Lanceta Gutiérrez [1,\*], Sara Perales-Momparler [2] and Miguel Rico Cortés [2]**

1   Faculty of Economics and Business Sciencies, International Doctoral School Researcher of the UNED, 28040 Madrid, Spain; alanceta3@alumno.uned.es (A.L.G.)
2   Green Blue Management S. L. (TYPSA Group), Paterna, 46980 Valencia, Spain; sara.perales@greenbluemanagement.com (S.P.-M.); miguel.rico@greenbluemanagement.com (M.R.C.)
\*   Correspondence: alanceta3@alumno.uned.es

**Abstract:** The Spanish Army is planning to create a new Logistics Base in Córdoba, in a development known as "Project BLET" (85 agricultural hectares, developed into an industrial zone). The Sustainability Concept proposed here is framed within the context of the strategic management process as a Strategic Business Unit. Aligned with 2030 AGENDA, the above mentioned process focuses on efficient water management and is providing a drainage strategy based on the use of Sustainable Drainage Systems (SuDSs), as well as a water management plan that aims to consider rainfall as a key non-potable water resource. The purpose is to prepare a roadmap, based on the Project, and a methodology that guarantees the viability of the SuDS and its non-potable rainwater use. The final result could be a sustainable military logistics hub in Southern Europe, which would stand out for being a pioneer in the treatment of rainwater and have sustainability features that are intended to be certified and assessed by different institutions.

**Keywords:** strategic project; strategic business unit; sustainable drainage systems (SuDS); nature-based solutions (NbS); sustainability certification; urban drainage

## 1. Introduction to the Army Logistics Base Project

The global security context has forced the Spanish Army to develop a strategy that is able to provide a technological evolution, in addition to addressing future challenges at the same time. Taking 2015 as the beginning of the period, the Defense Technology and Innovation Strategy [1], aligned with the R&D Ministry of Defense's efforts, seeks to develop military capabilities as a strategic objective. For this reason, the Army has the obligation and responsibility to prepare effective forces that are able to combat today and in the future. With this objective, in 2018, Spain began an ambitious project called "Army 35", which has to be developed in an interoperable and hyperconnected future scenario.

Within the "Industry 4.0" concept, BLET Project's functional needs would officially materialise on 13th February (2019), aligned with the innovation strategy [2]. From that time on, the "Technological Component" and "Requirements Definition" were formulated. Next, during 2020 [3], the essence of contributing to the development of military capabilities was continued. Finally, in 2023 [4], the most current materialised as axes not only the strategic autonomy increase (defence industry terms), as well as the Europe Defence contribution, but also the sustainable defence industry consolidation. This is what aligns with one of the main objectives of our research article: the implementation of SDGs.

The so-called "Army 35" [5] requires logistical support, which must be designed in accordance with the new procedures that are being developed. In this context, the future Army Logistics Base Project was born within this idea of adaption. This is a Project in which the Army seeks the rationalisation of logistics support infrastructure, while optimizing

logistics processes [6]. The Army Logistics Base Project (also known as "Project BLET" by its Spanish acronym) is the result of the integration of the twelve current logistics infrastructure army bases around national territory into a single facility. The planned location is in the outskirts of Cordoba, which is a city of 320,000 inhabitants located in the south of Spain. In this sense, within this city, a general action protocol would be signed on 17th September for construction and start-up [7]. The project consists of an 85-hectare base (Figure 1) with more than thirty military assets, which will be constructed on land currently classified as agricultural, as shown in the following figure.

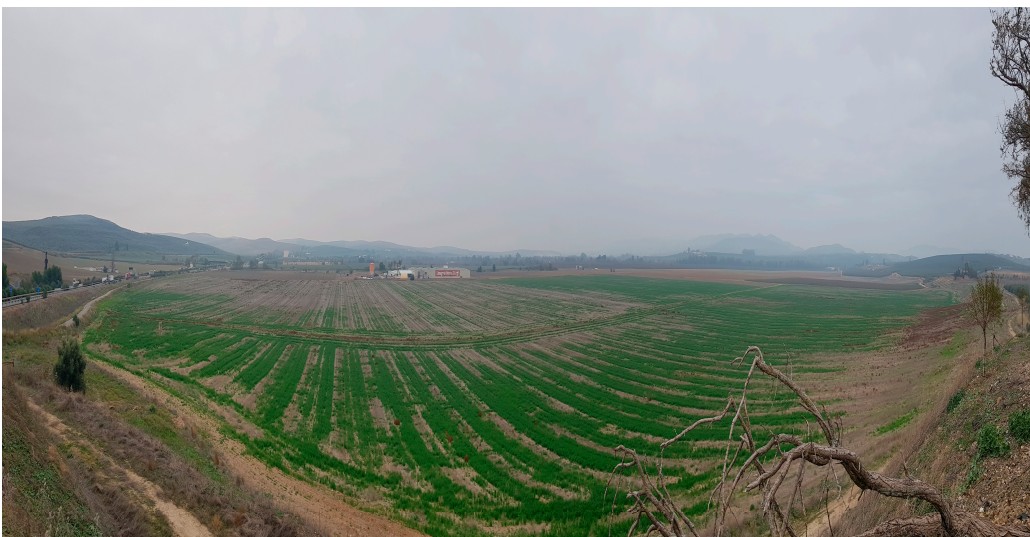

**Figure 1.** Pre-development scenario. Source: produced by the author [8].

The future Logistics Base is being developed as an ambitious, sustainable, and state-of-the-art technological project, whose core is high maintenance and supply level. In addition, the project aims to become a technological reference and a model to be followed in terms of sustainable infrastructure design [9] and, therefore, the future base faces the challenge of achieving minimised environmental impact.

The future BLET, which is expected to become a logistics hub in Southern Europe [10], will inevitably produce an environmental impact due to its industrial activity. Therefore, this impact needs to be analysed and consequently reduced or eliminated if possible. Specifically, due to the important industrial component it entails, it must be observed not to cause significant damage to the environment.

Considering the international commitment to the environment, in which Spain is also involved [11], new infrastructures must tend towards energy efficiency, developing, among others, a water management strategy, a deep analysis of the life cycle of materials, and a waste management plan. The present management model would contribute not only to the adaptation and mitigation of climate change but would also ensure that economic investment contributes substantially to the local and national circular economy. All of these approaches and considerations would constitute an important milestone to achieve the Sustainability Certifications.

Our research purpose is providing authority a methodology to follow, within a centralisation and new creation of infrastructure project, constituting a roadmap backed by objective calculations, adaptable to any geographical location. In this way, authority will have the capacity to carry out an analysis focused on some variables defined in this study, such as the amount of existing infrastructure, the impermeable surface caused, the vegetation, etc. Therefore, this will enable systems implementation and rainwater to be taken advantage of in a specific place, with determined conditions adaptable to any geographical location.

The aforementioned study will allow these systems to be designed in relation to the needs and will provide flexibility to the competent authority to make decisions, including storage and water use (all within the legal framework of international commitments, in which Spain is a participant, and the societal aim to be sustainable and help care for the planet).

## 2. Objectives

In general terms, the objectives associated with the purpose of this article are the following:

### 2.1. Strategic Objectives

(a)    Achievement of the Sustainable Development Goals (SDGs) established by United Nations.
(b)    Becoming a referent infrastructure in the implementation of SuDSs, reorienting the current urban model.

### 2.2. Specific Objectives

(a)    Development of a management model within which the water management strategy is sectorised as a Strategic Business Unit in order to carry out a Management Plan.
(b)    Alienation with the local governor.
(c)    Promotion of water-sensitive city design.
(d)    Control of contamination in relation to uses.
(e)    Take advantage of rainwater to satisfy the future Logistics Base non-potable water demand and return any excess to the subsoil, within the same conditions that nature would.
(f)    Achievement of a sustainability certification. The tools that legislation offers today are far below other standards offered by other independent evaluation systems.

## 3. Initial Hypothesis

In any scheme for the execution of infrastructure construction, both in the housing project and the building project, addressing the methodology according to the configuration of a Strategic Business Unit and considering the determining variables, we are able to affirm that the reuse, storage, and return of rainwater to the subsoil is possible in a specific location.

## 4. Determining Variables

(a)    New infrastructure geographic localisation.
(b)    Monthly collecting rainfall data.
(c)    Native vegetation and demand for irrigation.
(d)    Composition of the soil.
(e)    Workflow infrastructure.
(f)    Total vegetation area.
(g)    Water collection area.

## 5. Assessing Sustainability in Certification Schemes

Sustainability and quality terms are commonly linked, and normally analysis focuses on sustainability and considers economic, social, comfort, etc. aspects. It is common to talk about sustainability, when perhaps we should talk about quality [12]. It is, therefore, necessary to develop a database that combines criteria, requirements, and strategies that allow a comparative vision within the evaluation process.

Fortunately, nowadays there are numerous tools able to carry out holistic sustainability assessments. However, and referencing again to Cabellos García (2015) [12], the tools offered by legislation as control systems are not sufficient, falling short of quality standards established by other independent evaluation systems.

The main regulation document that currently fixes the quality standards for buildings in Spain is the Technical Building Code (also known as CTE by the Spanish acronym) [13]. The aforementioned Code, applicable to public and private buildings, establishes the

requirements that infrastructures should take into account basic safety and living requirements. Additionally, although it has been modified on issues like energy efficiency [14] (almost zero consumption), there are other certification systems that allow for the evaluation of the sustainability of a project in a more complete way, from different perspectives and standards, and provide a certificate according to the assessment obtained. Therefore, in addition to the aforementioned Technical Building Code, the recommendations extracted from international organisations and the different sustainability certifications will be taken into consideration for the project.

In the particular case of this study, the environmental analysis will consider the criteria set by VERDE certification (Valoración de Eficiencia de Referencia De Edificios), developed by Green Building Council España, the German DGNB certification (Deutsche Gesellschaft fur Nachhaltiges Bauen), and the British BREEAM certification (Building Research Establishment Environmental Assessment Methodology). Additionally, taking into account the location of the future Logistics Base in Cordoba, any aspect of the Spanish legislation with regards to the environment would be also added to the assessment.

## 6. Management Model. A SuDS as a Strategic Business Unit (SBU)

As a next step, the strategic management model must be developed, taking as a reference the process defined by Guerras and Navas [15], the Project Management Body of Knowledge [16] (PMBOK guide), the earned value analysis, and the lessons learnt from meetings with stakeholders, companies, and consultants.

To achieve the targeted sustainability certifications, it is necessary that the project follows a methodology based on a strategic direction that provides adequate guarantees for achieving the fixed objectives. To carry out the sustainability certification process, the project should focus on the definition proposed in 2006 by Johnson, Scholes, and Whittington [17], in which there is general agreement: "A strategy is defined as a long-term direction that allows achieving an advantage, in a changing environment, by configuring its resources and competencies, in order to satisfy the expectations of the interested parties". This is, precisely, the definition applied here, considered as "the search for a future strategic adjustment in a changing environment, linked to the business concept, in this case, in favour of the environment."

This objective would be achieved within the international legal framework to which Spain contributes, such as the United Nations Sustainable Development Goals [18], the legally binding Paris Agreement [19] reached at COP21, and the European Green Deal [20], the latter defining both the objective and the route to achieve them. All of them would lead to regulations, such as the EU Delegated Regulation 2020/852 [21], which implies the EU's progression towards an economy that serves people for environmental impacts. In this way, the European Green Deal would be the strategy to follow and the EU taxonomy would constitute the instrument that would facilitate the role of financial markets.

Therefore, management plans must be developed in accordance with the required sustainability certification. Specifically, in terms of water optimisation, a water management plan (Figure 2) must be prepared considering, in our case, Sustainable Urban Drainage Systems as a Strategic Business Unit (SBU) [22]. This is a fundamental part of the implementation of the sustainability strategy, favouring independent activities and objectives, but always related to the overall objective of the Project. Professionals and researchers had the opportunity to take advantage of this sectorisation to delimit business risk as a systematic and organised innovation activity for strategic renewal [23].

Thus, based on a methodology that lays the foundations for success (70% of projects without methodology fail [24]), a general path is created for the definition of the requirements that will allow us to achieve the sustainability certification and, in particular, to define the actions that materialise these requirements and objectives in a contractual manner for the different bidding specifications.

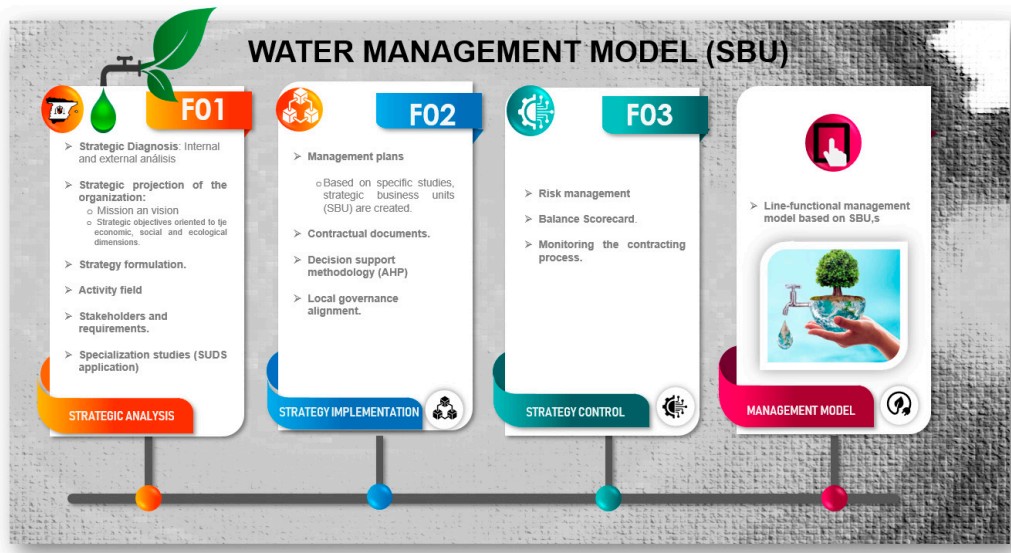

**Figure 2.** Water management model development within Project BLET. Source: produced by the author.

In this way, and focusing on urban water, we propose a water management plan that satisfies the requirements of the conceptual and design stage, and Sustainable Drainage Systems (SuDSs) appear as an alternative approach to conventional drainage systems (strategic analysis). In the case of locations with significant water stress, such as the city of Cordoba, these systems are especially relevant in order to improve water management during both flood and drought episodes. It is important to highlight that SuDSs are not being used in Cordoba currently and, therefore, this proposal would be very innovative, and it could become an important landmark for the city. In fact, the Agreement signed with Junta de Andalucía on 27 January 2023 [25], where the financing and execution of investments are ordered, the proposed solution included a sustainability certification considering the use of phase water consumption as a basic level indicator.

## 7. Review of the Literature

There is a large amount of historical precedents related to rainwater capture and reuse, especially in areas with a high water stress component.

As different studies attest [26], rainwater meets most of the quality indicator parameters and can be used for various purposes, such as garden irrigation. Systems that use effective rainwater management dates back to communities such as the Roman Empire or the Mayan Empire. However, its implementation is very scarce [27]. In fact, nowadays, at the geographical location referred to in this research article, the implementation of sustainable drainage systems is pioneering.

It was during 2013 (United Kingdom) when CIRIA (Construction Industry Research and Information Association), defined SuDSs as "those drainage systems that contribute to sustainable development and the improvement of urban design, balancing the different interests that influence the development of the community". They focused on the management of surface water, considering the quantity of water (floods), the quality (pollution), and the public use that can be given to it [28], defending the use of SuDSs as a key measures.

However, there are authors who approach the rainwater reuse in a different way, such as Polledo Pereda [29]. His study on the "recovery and reuse of rainwater" refers, since 2019, to several projects that allowed rainwater to be collected for different purposes, such as washing, toilets, or washing industrial vehicles. Mr. Polledo starts from rainfall data (middle value) and observes the waterproofed roof area. Subsequently, he plans accumulation tanks and uses them for sports facilities, orchards, and gardens, or washing.

It was during 2019 when the Spanish Ministry for the Ecological Transition addressed a range of SuDS types, as well as storage systems, and included the permeability coefficient. This methodology consisted of a classification of the Iberian Peninsula into three climatic zones, providing an average annual rainfall for each of them. Based on this structure, the design of the infiltration and storage elements proceeded [30].

Other authors, such as Ahsen Maqsoom during 2021 [31], introduced the BIM model for the design and calculation of estimated data, such as the roofs area, to empirically calculate the rainwater storage potential (it is argued that BIM modelling provides the theoretical basis for identifying rainwater capacity). The aforementioned authors used the annual accumulated precipitation of the study sites as data, obtaining average values of rainwater accumulation, even performing a monthly balance calculation between demand and available water.

An example for the defence of SuDS use can be found during 2022 in the Provincial Council of Granada [32]. Its Project for the use of rainwater in municipal buildings and facilities aimed to use water collection systems that allowed the collection of all the rainwater that usually ends up as runoff and proposed the SuDS as a collection system, returning the surplus water to the natural environment. The Project proposed the most adapted typology to potential areas, but it did not examine in detail size or water collection data, estimating an average of the Granada municipalities.

The research that led to this article will defend the advantages of SuDSs, but more efficiently and accurately.

## 8. Introduction to Sustainable Drainage

The use of Sustainable Drainage Systems (SuDSs) has grown considerably in Spain over the last decade, and they are starting to be considered as a key element to provide flood resilience to new developments, in addition to managing stormwater in a sustainable way [33].

The loss of the soil permeability associated with urban development is increasing the runoff rates discharged to water courses, which can lead to flooding issues further downstream. The agricultural area replaced by infrastructure makes a significant impact on the runoff, disturbing the infiltration capacity and increasing flooding frequency, as cited by the Mexican Geological Sciences magazine in "the spatio-temporal change of use soil and its relationship with flooding study" [34].

Additionally, water quality can be compromised by the pollutants that are swept up by the runoff on its way to the conveyance network. In essence, a drainage strategy based on the use of SuDSs can help to prevent these problems while providing additional resilience in the context of climate change, where the frequency and intensity of extreme rainfall events are increasing.

In this context, SuDSs emerge as a solution to replicate natural hydrological processes, avoiding the negative impacts on the water cycle caused by urban development. SuDSs aim to reverse the negative effects derived from urbanisation processes, increasing soil permeability and significantly reducing peak flows, as stated by Ortuño in studies carried out in Alicante (Spain) [35].

The different types of SuDS features can suit many different design scenarios and provide benefits from multiple perspectives. From the hydrological point of view, SuDSs are designed to reduce the runoff volumes discharged, infiltrating part of this runoff; to improve water quality, taking advantage, for example, of the filtration through granular media; and to attenuate flow rates, providing temporary storage in order to prevent downstream surcharge or flooding. Among these benefits, SuDSs also allow the temporary storage of part of the runoff that is not infiltrated in the ground, in order to enable its use for compatible non-potable uses, such as irrigation, vehicle washing, or flushing (several studies have reported the presence of pathogens in runoff water [36]).

In this way, following a methodology within Project BLET, an effective strategy towards water efficiency would be designed.

The sustainable drainage approach is supported by Spanish regulations in terms of urban water. Particularly, the recently issued Real Decreto 665/2023 [37] provides a consistent definition for the SuDS, and gives an indication of the main design objectives in terms of runoff quantity and quality. On the other hand, the current proposal of the Directive of the European Parliament and of the Council concerning Urban Wastewater Treatment [38] focuses on the importance of green developments and the hazard of the potential pollutants captured by the urban runoff.

## 9. Project Context

As described previously, the BLET will be constructed in Cordoba, in the southern half of Spain. The factors that have determined the geographical location of the logistics technology centre in the city of Cordoba have been, among others, the geostrategic position [39].

In order to develop an operational solution that includes the concept of sustainability for the future facility, and the definition of the different scenarios for action, it is essential to align with local governance. With the aim of integrating the Logistics Base within the city's Strategy, it is crucial to involve the local government as an important actor [40], advocating for greater empirical, social, and institutional research and betting on transnational municipal networks (TMN) as an instrument for climate change mitigation at a local scale. The intention, as defended by the authors Rodríguez Abarca and Niño [41], is to rethink current urban models, improving urban comfort and making cities more resilient to climate change. All of this, besides the studies carried out by Wong and Brown [42], urges cities to take advantage of different sources of water supply. It is necessary to offset the demand with the use of alternative and sustainable energies.

Cordoba has a typical mediterranean climate (Csa, according to Köppen climate classification), with warm summers and soft winters. It is located within the B4 climate zone. Rainfall is irregularly distributed along the year, with an average annual volume of 600 mm.

With regards to local stakeholders, urban water is managed by EMACSA (Empresa Municipal de Aguas de Córdoba, Sociedad Anónima), Cordoba's local water authority. In 2012, this institution issued a design manual that provides specific information about drainage network design, including an IDF curve, obtained from local observed data between 1984 and 2012. In terms of design return periods, the manual recommends developing drainage infrastructure able to deal with the 25-year return period events with no flooding (highly populated areas [43]).

In order to complement the information provided in EMACSA's design manual, as part of the drainage strategy works, an analysis of the available rainfall data for the nearest rain gauges nearest to our location may be realised (the closer the value of each location, the higher optimisation level). This analysis includes the estimation of daily rainfall percentiles for the selected rain gauges, which can provide useful information about the regular rainfall regime. To summarise the results obtained for the different rain gauges specifically on BLET's possible location, the Inverse Distance Weighted methodology may be applied [44], achieving the results shown in Table 1.

**Table 1.** Obtained rainfall percentiles after applying the Inverse Distance Weighted method.

| Percentile | Daily Rainfall (mm) |
|---|---|
| $V_{95}$ | 35.1 |
| $V_{90}$ | 26.3 |
| $V_{80}$ | 17.6 |
| $V_{60}$ | 9.3 |
| $V_{50}$ | 7.1 |

On the other hand, geotechnics can play an important role in the drainage strategy. BLET's proposed location sits on top of clay and lime soil stratum, with a gravel stratum

underneath this layer. Clay and lime soils tend to be quite impermeable, while gravel soils typically have higher infiltration rates. It is likely to find a mixture between these soils and, therefore, the actual infiltration rate can vary significantly within BLET's boundary, depending on the clay content in each particular location. The drainage strategy must take these circumstances into consideration, and it must aim to provide versatile solutions that can be easily adapted once further permeability information is available.

In order to know the terrain and more effectively approach the SuDS design and implementation, it is necessary to know the permeability coefficient and terrain composition. This will allow to carry out detailed hydraulic modelling, defined below. Both types of data can be extracted from the geotechnical study of the area or, failing that, follow the methodology developed by different publications, such as the one developed by the author Ruiz Vargas in May 2017 [45].

To do this, prospecting explorations will be carried out, consisting of cone penetration tests, which can alter samples of the terrain obtained. The pits are presented as the most used prospecting method (excavation of vertical wells that reach depths of three or more meters).

The trenches should be distributed homogeneously throughout the entire plot, with more incidence in those places that are susceptible to the SuDS implementation.

## 10. Methodology and Design Criteria

Cordoba has not developed any drainage strategy based on the use of SuDSs to date and, therefore, some uncertainties may appear [46], which may cause physical, perception/information, and organizational barriers. The factors that set the guidelines for defining the design criteria in the conception of the storm drainage strategy have been stated, establishing, in this case study, the future BLET as a reference in military installations.

The future Logistics Base must be aligned with local entities by seeking and adopting strategies that are developed in favour of sustainability, in general, and in favour of adequate management of stormwater, in particular. In this sense, the City Council of Cordoba has developed its AGENDA CÓRDOBA [47], committing to the optimisation of water management, as well as the reduction of the impact of flooding due to the effects of climate change. This Agenda pursues the fulfilment of Sustainable Development Goal number 6 of the United Nations Organization "Clean water and sanitation" and links it with Objective number 4 of the Spanish Urban Agenda, which establishes the obligation to carry out management sustainability of resources, promoting the circular economy, optimising and reducing water consumption.

The location of the future BLET will alter the natural hydrological cycle, with a loss of vegetation and waterproofing to the current agricultural land, which will presumably lead to an increase in runoff volumes. Sustainable Drainage Systems (SuDSs) are presented as complementary systems to traditional ones that base their purpose on replicating the natural hydrological process, acting at the source of runoff before the flow becomes a problem, infiltrating, storing, or evapotranspirating as much rainwater as possible.

The drainage strategy has been planned from a holistic perspective as an effective alternative to counteract these negative effects, aiming to maximise the benefits from a sustainable drainage approach. This methodology was synthetised by The SuDS Manual [48] in four pillars, which represent each of the different aspects that a well planned SuDS design should take into consideration:

### 10.1. Designing for Runoff Quantity

Firstly, the amount of runoff produced within the site and its variation from the baseline scenario must be analysed (Figure 3). In order to preserve the current condition, and to avoid increasing the flood risk of downstream assets, as a design criterion, it can be stablished that the peak flow to be discharged out of the development boundary should be kept below the pre-development runoff rate for 25-year rainfall events. It is important to highlight that this criterion is aligned with the spirit of the new Real Decreto 665/2023,

which states that new developments must include SuDSs able to avoid increasing flood risk downstream.

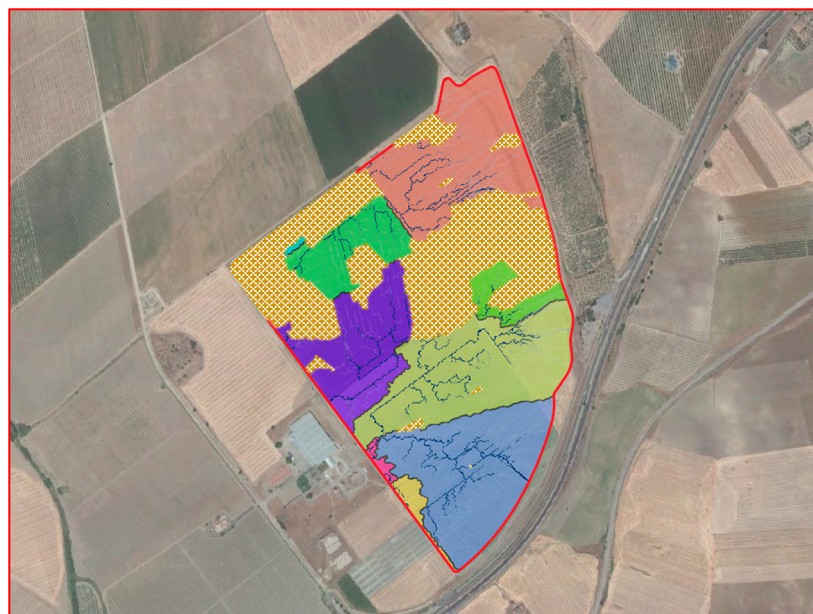

**Figure 3.** Pre-development catchments within BLET's boundary.

This pre-development runoff rate is estimated using the procedure stated by the Spanish road drainage design manual, which consists of a variation to the Rational Method including some additional parameters. Thus, it could be concluded that the peak flow discharged in the post-development scenario should not exceed 2.02 m$^3$/s, providing the required attenuation storage to meet this requirement.

Moreover, preserving the existing hydrology and protecting downstream assets, it is important to consider eventual changes on the rainfall patterns caused by climate change. Therefore, following the recommendations from CEDEX [49], a climate change increasing factor of 15% may be applied to the rainfall intensities contained on the IDF curve provided by EMACSA [50], as shown in the following table:

Cordoba commonly experiences droughts and, hence, runoff shall be also considered as a water source. Therefore, the water resources strategy will focus on collecting as much rainfall as possible in order to store it temporarily, and keep it available to use it in usages compatible with non-potable water, such as watering gardens or cleaning processes.

*10.2. Designing for Runoff Quality*

As well as considering the runoff quantity, drainage strategy the importance of water quality for the downstream aquatic ecosystems must also take into consideration. Therefore, runoff will be provided with an appropriated level of treatment before getting infiltrated or being conveyed out of the site. This treatment is enhanced by using SuDSs, and characteristics and pollutants mitigation capacity should be sufficient considering its catchment land use.

In order to select the right type of SuDS in each catchment, the Mitigation Index (The SuDS Manual, CIRIA) approach has been used. This methodology assigns different pollution hazard indices for the most common pollutants considered (suspended solids, metals, and hydrocarbons) for different land use classifications. These pollution hazard indices must be compared with the mitigation index of the different SuDS technics, which are also established in this methodology based on their treatment potential.

### 10.3. Designing for Biodiversity

It is acknowledged that the SuDS based on Green Infrastructure, such as bioretention areas, swales, or ponds, have a higher impact on biodiversity than other types of underground SuDSs. Therefore, the drainage strategy should prioritise the inclusion of these green above-ground SuDS features, which will maximise the benefits of their inclusion and can be easily aligned with garden plans.

### 10.4. Designing for Amenity and Public Utility

Given the nature of the development and its future use, it is crucial to understand which spaces are available to include SuDSs and which areas must be kept clear for future operations. Gardens, plazas, or landscape verges become an opportunity to include SuDSs that not only contribute to managing runoff at source, but can also turn into an attraction themselves, allowing alternative uses during dry weather.

## 11. Drainage Strategy Overview

Once the design principles have been fixed, the drainage strategy must be developed considering specific site constraints. Firstly, internal plots could be carefully analysed, finding opportunity spaces that allow the integration of SuDS features as part of the landscape design [51]. As mentioned earlier, this process must take into consideration future operations of the development, and SuDSs will be proposed in locations where they should not have an impact on the normal activities, typically gardens or verges adjacent to roads.

Next, the SuDS type selection process has to be done, taking into consideration the treatment potential and land uses across each catchment, following the Mitigation Index methodology. The shape and amount of space available should also be taken into account in this process, in which generally SuDSs based on Green Infrastructure are generally prioritised due to their ease of integration as part of the landscape design and their biodiversity benefits.

Finally, different SuDS features would be individually sizes to ensure they are capable in treating a reasonable amount of runoff, assuming they will overtop the conveyance storm network when their nominal capacity is exceeded. For this purpose, the rainfall percentiles methodology must be used, which allows the easy estimation of the volume and dimensions required for each SuDS to store the amount of runoff associated with a certain percentile of daily rainfall. As a general rule, SuDSs are designed to fully treat the runoff associated with the $V_{95}$ percentile (which has been estimated for this location in 35.1 mm), which states a daily rainfall volume that should not exceed 95% of the rainy days [52]. In cases where site constraints do not meet this requirement, this target volume is reduced to the $V_{80}$ (estimated in 17.6 mm), which is a typical reference threshold in many international and national recommendations, including the Real Decreto 665/2023.

$$V_{req} = V_x \, A$$

Calculations developed, obtained the volume of each SUDS from their area and depth, assessing then if the volume is sufficient to support 95% or 80% percentiles to be able to carry out the Project requirements. It is essential to carry out controlled drilling and a geotechnical study to determine the permeability of the land. Once the study is completed, the areas that allow infiltration can be determined and the SuDS can be optimally designed, adapting it to the required volume and depth.

The following figure (Figure 4) shows the SuDS types proposed in the different plots within the BLET (restricts building on the section). The proposed nature of the scheme is emphasised (the contracting authority will be the one to decide with the awarded company, and the certification function).

An initial assessment of the proposed SuDS typology must be made based on variables of the technique and the type of management that will be carried out with water [53].

In general, verges adjacent to roads could be used to include Bioretention Areas, where treatment potential is enough to receive runoff from trafficked areas. This type of SuDS (in green in the figure below) should also be used to drain areas where heavy or military vehicles may park or drive. Therefore, it is necessary to include these areas where heavy metal contamination in stormwater runoff is greater.

On the other hand, carparks for private vehicles are proposed to include Permeable Paving in parking spaces, which should be able to capture and infiltrate the runoff produced on their own surface (in yellow in the figure below).

Proposed landscaped areas within the plots may include Rain Gardens (in blue in the figure below). They can superficially store runoff that filters through a certain medium, such as topsoil and Detention/Infiltration Basins (in red in the figure below), which take advantage of terrain depressions in addition to favouring biodiversity and allowing flow runoff through a controlled drainage, which are designed to receive runoff with a lower risk of pollution, typically from pedestrian areas, footways, gardens, or building roofs. These open SuDSs can be easily integrated as part of landscape features. Instead, we could consider Harvesting Systems for the largest warehouses, which could include the largest roofs (in orange in the figure below). Finally, some of the hardstanding areas should be provided with geo-cellular storage tanks as they must be kept clear for the normal operations of the site and do not allow the inclusion of open SuDS.

Although all the SuDSs are individually designed to provide a sufficient treatment volume, this condition does not ensure that the overall discharge rate does not exceed the pre-development runoff rate as established in the design criteria. In order to test the performance of SuDS systems under EMACSA's design storms, which may cause overflow to the storm network, a detailed hydrological–hydraulic model may be built. This modelling phase would allow the overall behaviour of the SuDS features included on the system to be checked, as well as the development of a first approach to the design of the conveyance storm networks required to collect overflows and underdrains (when required) of the SuDS proposed across the site.

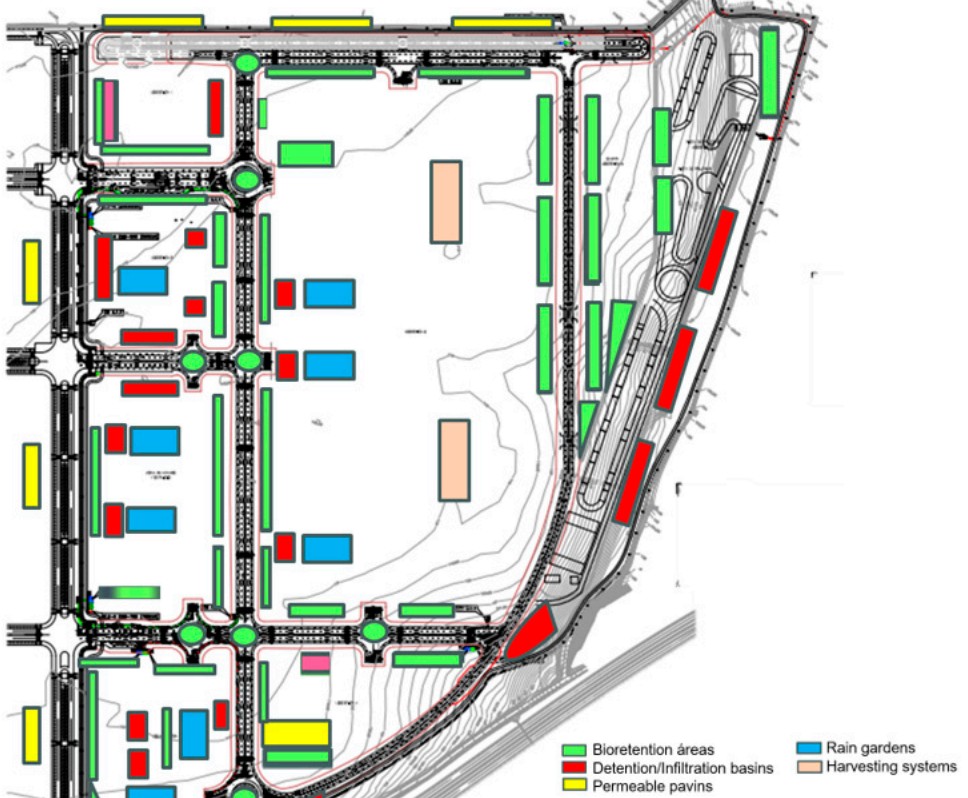

**Figure 4.** Proposed Drainage Strategy SuDS distribution within BLET.

## 12. Detailed Hydraulic Modelling: Scenarios and Simulation Results

To carry out detailed hydrological–hydraulic modelling, software specialised in sustainable drainage may be used (for example, in our case, InfoDrainage Autodesk).

There are other hydraulic calculation models on the market that have proven to be effective as an analysis tool, but we selected the Autodesk InfoDrainage hydraulic motor. Info-Drainage Autodesk allows you to transform CAD models into BIM and easily integrates with bidirectional compatibility with Civil 3D, providing a realistic 3D view.

The software uses traditional hydraulic and hydrological equations (manning, dynamic wave, accumulation, and washing of contaminants, etc.) to accurately simulate the behaviour of runoff in urban basins. In addition, the tool allows each of the proposed SuDSs to be defined in detail, so that specific project conditions can be reproduced. Additionally, the program allows good integration of CAD tools, which allows information to be imported and exported efficiently, and contributes to generating models that adequately represent the actual shape of the proposal.

To build the model, different elements that were part of the drainage strategy were introduced through the software interface. Essentially, the model was formed by different catchments, SuDS features, pipes, and manholes, which have a large number of parameters (dimensions, elevations, porosities, etc.) to guarantee that their hydraulic behaviour matches the finally constructed element. To ensure that the geometry of the SuDS and the longitudinal profiles of the pipe networks were realistic and site specific, a Digital Terrain Model (DTM) was also introduced in the software environment.

Once the model was constructed, the simulations scenarios were set. The proposed drainage strategy was tested under seven different storm durations, from 10 min (higher intensity) to 720 min (higher volume), considering a climate change allowance of 15%, as shown in Table 2. Additionally, three different geotechnical scenarios were considered, with various infiltration rates fixed, which aimed to reproduce the uncertainty expected in terms of soil permeability.

**Table 2.** IDF curve provided by EMACSA and IDF obtained after applying a climate change increasing factor.

|  | **Duration** | **10** | **30** | **60** | **120** | **360** |
|---|---|---|---|---|---|---|
| IDF-EMACSA | Intensity (mm/h) | 113.9 | 56.3 | 46.8 | 31.4 | 15.8 |
|  | Rainfall (mm) | 18.1 | 27.6 | 47.2 | 64.2 | 92.8 |
| IDF-EMACSA + 15% climate change | Intensity (mm/h) | 130.9 | 64.7 | 53.8 | 36.1 | 18.1 |
|  | Rainfall (mm) | 20.8 | 31.7 | 54.2 | 73.8 | 106.7 |

The simulation results showed a good performance of the system for all the scenarios considered. As expected, the performance of the different SuDSs varies depending on the duration of the simulated rainfall and the infiltration scenario being analysed. For example, in the case of the Bioretention Areas or the Rain Gardens, the model shows that, for low volume rainfall, the runoff reaching ponding areas was temporarily stored on the surface for later percolate through the filter medium, without overflowing the storm network. On the other hand, if the storm volume exceeded the capacity of the ponding area, part of the runoff flowed through the overflow system directly to the storm network, as reflected by the hydrographs of Figure 5.

Figure 5 contemplates two different scenarios for a bioretention area that has been taken as an example: one with an input water flow of 60 minutes duration and another one of 360 minutes duration. During the first, the SUD perfectly absorbs the runoff, without overflow occurring, facilitating the outflow through the drainage pipes. In the second case, an overflow occurs that would cause the excess water to be diverted directly to the drainage network.

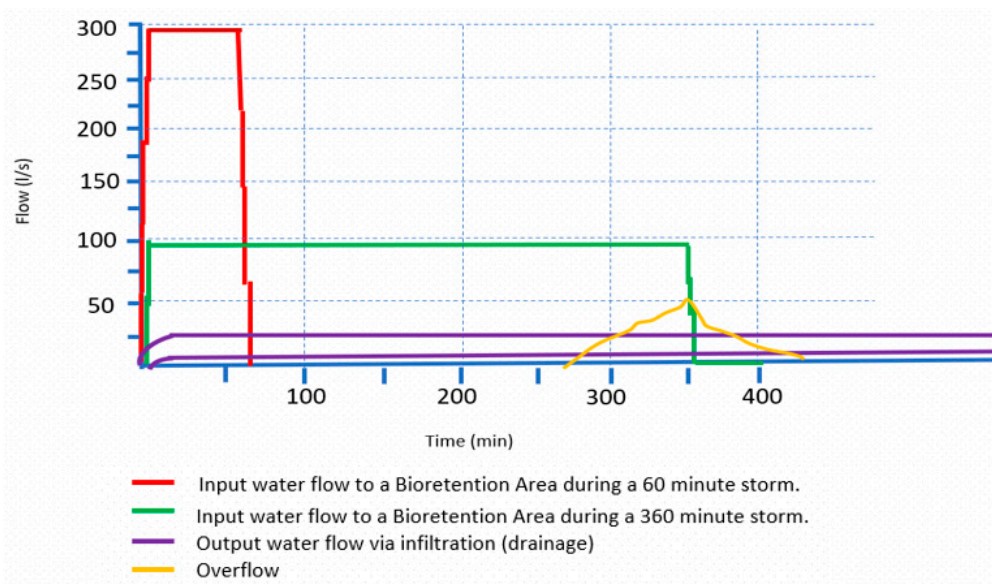

**Figure 5.** Hydrographs for a Bioretention Area performing with "overflow"and "no overflow" for a 60 and 360 min duration storm.

As previously mentioned, one of the main objectives of the modelling was to evaluate if the proposed SuDS allowed the restriction of flow discharged out of the BLET to the pre-development runoff rate, estimated in 2.02 m$^3$/s.

To add some context to the numbers below, the last row of the Table 3, includes an additional simulation, which corresponds to an alternative drainage strategy with no SuDS or attenuation storage, where the runoff is captured and conveyed out of the site via a conventional piped network. The discharge rate obtained is 18.98 m$^3$/s and, hence, the discharge rate reduction obtained with the proposed drainage strategy could be about 80%.

**Table 3.** Overall discharge rates extracted from the hydraulic model for the different design scenarios.

| Design Scenario | Maximum Discharge Rate (m$^3$/s) |
|---|---|
| No infiltration | 2.02 |
| Alternative drainage strategy No SuDS | 18.98 |

## 13. Water Management Plan Based on the Strategic Business Unit (SBU)

The importance of water as a resource in an area with a significant water stress, such as the city of Cordoba, requires the definition of a strategy for maximum use of rainwater. Rainwater can be collected, stored, and used as required (for compatible non-potable water usages), reinfiltrating into the ground any eventual exceedance volume, in the same conditions that nature would do.

Water demand and its availability must be analysed carefully in order to provide appropriated harvesting storage volumes.

The calculation methodology involves compensating the general demand with the captured rainwater, all under the prism of total available water. The success of the strategy is the balance between the water demanded and the water recovered, establishing the future BLET as a sustainable infrastructure in water use terms.

In this way, water needs and how to satisfy the consumption would be analysed, considering the following parameters:

- Water availability for future BLET, located in The Rinconada industrial city (Cordoba).
- Human use water demand (non-potable consumption): sanitation and demand for buildings, including leisure time facilities, and demand for irrigation.

- Industrial use water demand.

  Additionally, it is assumed that some preventive measures will be installed, such as:
- Sanitary appliances devices that force low water consumption.
- Use of management systems that are capable of detecting water leaks, thanks to the use of sensors and monitoring.
- Favouring the native biodiversity of the area, with special emphasis on low water needs plants, in order to reduce irrigation demand.

  The set of the proposed parameters and the stated preventive measures will allow a balance between demand, available water, and harvesting. As part of this balance, the SuDS will collect, treat, and store runoff for future use where required.

### 13.1. Buildings' Roof: Collectable Rainwater

Runoff captured directly from the building's roof is considered optimal in terms of water quality and ease of collection, and it can be a significant water source, for example, for irrigation. The analysis, which is based on parameters defined by different sustainability certifications, takes the usable roof area ($m^2$) of each infrastructure and its runoff coefficient. This is known as "surrounding conditions".

Taking the Building Project information published in the Spanish State Contracting Platform [54] and other publications as Boletín Oficial del Estado [55], the collection capacity of each building is estimated on a monthly basis (Table 4), resulting in an annual volume of collectable rainfall ($m^3$). For example, the first row of the following table illustrates the calculations for the heavy industrial warehouse (main supplying area), with a surface area of approximately 50,000 $m^2$, and the collectable stormwater volume for each month. If the same methodology is applied to the rest of the planned buildings, the overall volume available for the BLET can be obtained (Polígono de la Rinconada's weather station [56], and the inverse distance-weighted interpolation method among different rain gauges provides the amount of rainfall in the area).

**Table 4.** Estimated date of facilities based on the VERDE sustainability certification and Polígono Rinconada's weather station.

| Typology | Jan. | Feb. | Mar. | Ap. | May | Jun. | Jul. | Aug. | Sep. | Oct. | Nov. | Dec. |
|---|---|---|---|---|---|---|---|---|---|---|---|---|
| Example: Heavy industry warehouse (m$^3$) | 2500 | 1900 | 1800 | 1900 | 1500 | 550 | 400 | 170 | 1200 | 3000 | 2900 | 4100 |
| Total BLET(m$^3$) | 8000 | 6500 | 6000 | 7200 | 5300 | 1500 | 240 | 600 | 4000 | 10,000 | 9000 | 14,200 |
| Estimated total volume of collectable water (all buildings) (m$^3$): 72,540 | | | | | | | | | | | | |

Considering a 20% loss (included in the final result), based on these data, it would be possible to estimate the water demand for different non-potable uses: industrial use, irrigation, and building consumption.

### 13.2. Necessary Water for Industrial Use

To define a ratio of water consumption per hour, the number and characteristics of washing facilities of the Project BLET were estimated. The data provided by the Spanish Health Ministry [57] and self-experience were used as a correlation, resulting in an expected volume of approximately 10,500 $m^3$ per year. The estimate would include vehicle washing, as well as assembly and subassembly washing (these activities are carried out daily).

### 13.3. Water Demand for Irrigation

Irrigation is expected to be one of the most significant water demands that can be served with non-potable water. However, it is necessary to favour the native biodiversity development (required by the terms of reference [58]) to reduce the "heat-island" effects (typically associated with cities). In order to be able to estimate the irrigation needs,

we could identify species and recommend swapping those that may require a higher water demand.

The main objective of the methodology is to coordinate the satisfaction of the hydrozones water demand, know the irrigation requirement curve, and try to match water collection during appropriate months. In calculations, we must consider local regulations (local governance alignment: Cordoba is an integrated region in Andalousia Autonomous Community). Thus, we can observe the species reflected in the "Junta de Andalucía Manual of Irrigation and Gardens" [59], obtaining their species coefficient (ks) (this coefficient is used to calculate the water according to the particular needs of each botanical species, using values between 0.1 and 0.9), and discarding those that have a higher coefficient ks, due to their high water demand or those that are not native to the area. Species that are autochthonous are shown in the "Anthos list of the Spanish Government" [60].

Therefore, by choosing species with low ks and native species and establishing them by a number of hydrozones, the irrigation needs may be defined, taking into account rainfall, evapotranspiration, hydrozone surface, and type of vegetation. These irrigation demands may be calculated on the basis of mentioned reference certifications, which consider both the aforementioned coefficient and the type of vegetation. On this basis, species such as the cork tree (quercus suber) can be proposed.

The green area foreseen for the BLET according to the Urbanization Project [61] is estimated to be 270,500 m$^2$ (it is based on irrigation system and gardening annexes). In order to optimise the water management, species must be grouped by hydrozones (Zero Consulting methodology [62], exhibited in the framework of "III Congreso de energías alternativas de la Defensa" [9]), so that plants with similar water needs are close to each other (Table 5). The methodology indicates the selection of similar species by hydrozones that will provide annual water needs.

**Table 5.** Estimated description of the surface of the hydrozones and their annual non-potable water needs.

| Hidrozone | Description | Estimated Surface (m$^2$) | Annual Necessary Water (m$^3$) |
|:---:|:---:|:---:|:---:|
| Z1 | Plots and tree pits | 150,000 | 19,000 |
| Z2 | Roundbounds | 4500 | 1000 |
| Z3 | Covers | 30,000 | 4500 |
| Z4 | Roads | 86,000 | 14,800 |
| Total volume of annual irrigation consumption (m$^3$): 38,500 | | | |

### 13.4. Necessary Water for Building Consumption

In order to estimate the non-potable water demand within buildings, the type of load (with a frequency of its sanitary appliances) and the typology of each building must be taken into account. Frequency was based on DGNB certification ratios and consumption of sanitary appliances on BREEAM certification [63].

Thus, in absolute terms, with calculations adapted to each building of the Project BLET, a consumption in buildings of 16,200 m$^3$ size per year was estimated.

In addition, there are also facilities dedicated to leisure time, such as swimming pools [64], which must be taken into consideration.

### 13.5. Demand Versus Rainwater Collection

Considering the total non-potable water demand of the future Logistics Base, which would include buildings (sanitary ware), irrigation, and industrial use as the main demand references, the annual volume of water required would be approximately 65,200 m$^3$ (Table 6). It must be said that the concept of "street washing" has also been included in the consumption of buildings (an annual ratio of water consumption per m$^2$ of road is obtained).

**Table 6.** Estimated description of the annual BLET non-potable water need.

| Water Volume Demanded (Annual) | |
| --- | --- |
| Origin | Volume (m$^3$) |
| Buildings (sanitary ware) | 16,200 |
| Irrigation | 38,500 |
| Industrial | 10,500 |
| Total (m$^3$) | 65,200 |

From the estimated demand data, it is feasible to develop a "reuse criteria", considering rainwater harvesting, grey water, other sources, and percentage diverted to SuDS. Therefore, it is possible to calculate the percentage of rainwater that would be annually stored for reuse and the part that will be infiltrated through the proposed SuDS, if the geotechnical conditions allow, or discharged through the stormwater network.

In addition to rainwater harvesting, there are other potential sources of non-potable water, such as the grey water. These alternative sources provide a greater resiliency to the whole system.

The term "grey water" refers to untreated household wastewater, which has not been contaminated by toilet waste. This water can be also reused for toilet flushing and cleaning if it is treated appropriately, becoming an interesting option for buildings with showers (gymnasium, temporary accommodation, etc.).

Additionally, some water usages allow for reuse ("other sources"), becoming an alternative source of non-potable water that can be considered in the analysis. This is the case, for example, in vehicle washing, where the eventual effluents can be captured and treated so it can be safely used again.

The Table 7 shows that demand is comfortably met with proposed alternative sources, offering an available ratio of water consumed/water. In this sense, it can be concluded that, thanks to a meticulous SuDS design, it is feasible to meet the Base's annual water demand, reinfiltrating any exceedance volume and contributing, through water recycling, to conservation and environmental protection.

**Table 7.** BLET water reused volume proposal.

| Water Reused Volume (Annual) | | | | |
| --- | --- | --- | --- | --- |
| Origen | Annual Volume (m$^3$) | Percentage Diverted to SuDS | Reinfiltrated Volume (m$^3$) | Reused Volume (m$^3$) |
| Rainwater harvesting | 72,540 | 20% | 14,508 | 58,032 |
| Grey water | 7150 | - | - | 7150 |
| Total (m$^3$) | | | | 65,182 |

## 14. Conclusions

The design phase of the new military logistic hub, known as Project BLET, is considering sustainability as one of the main inputs [65]. The development will be constructed in Cordoba, and involves an area of eighty-five hectares and more than thirty buildings, becoming an important milestone for the Spanish Army and for Cordoba city.

The raised design process in this article follows a methodology that, with an initial strategic analysis, would allow the implementation of a strategy based on the objectives set. This is how the term "strategic business unit" is born, which, particularly applied to the Water Management Plan, would allow the development of a sustainable drainage strategy and water resources plan, in accordance with the fixed objectives.

The proposed drainage strategy allows a solution based on different Sustainable Drainage Systems (SuDSs) across the BLET, which would be aimed at infiltrating or attenuating the runoff captured in order to achieve the design criteria proposed, which are focused on preserving the pre-development hydrologic and protecting water courses from

runoff pollutants, following the spirit of the recently issued national regulations. A detailed hydraulic model has been prepared in order to check that all design conditions are met for the different geotechnical and rainfall scenarios considered.

On the other hand, the Water Management Plan aims to consider rainfall as a water resource for non-potable usages, such as irrigation, industrial washing, or flushing. The estimated calculations carried out prove that the collected annual rainfall volume on main roof buildings (in addition to other sources of non-potable water) should be sufficient to supply the non-potable water demand of the BLET, which has been estimated according to local and national manuals.

Thanks to the implementation of the SuDS, collected rainwater satisfies the demand for non-potable water of the Logistics Base, and reinfiltrates into the ground the percentage that is determined, under the same conditions that Nature would do. The new and huge planned infrastructure is capable of supporting the demand for non-potable water, considering the proposed systems.

The methodology, in this case, provides a study tour in which the location of the new installation is first analysed. Based on official organisations, data in the area, and the Inverse Distance Weighted methodology, we are able to know the amount of rainwater that is generated on a monthly basis. This will allow us to make accumulation or reinfiltration decisions considering the demand for irrigation of green surfaces and, in addition, compensate for the greatest demand for non-potable water of any installation. In this sense, we have described that annual rainfall is not addressed in a general way or are generic classifications taken into account.

So, in a precise and a detailed way, it is necessary to know the surface that is capable of collecting rainwater and, depending on its nature and location, to design an adequate SuDS that allows both storage and infiltration (at the same time, operational flows of the Base are not interrupted).

In order to optimise design, factors such as the soil depression (retention ponds), large roof surfaces (harvesting systems), or promoting biodiversity (rain gardens) will be taken into account. However, as defended by different authors, a preferred system design has not been found to maximise performance [66].

On the other hand, it has been analysed that the demand for irrigation represents the most important value. Therefore, it is essential to consider its effects. The creation of green spaces represents not only comfort and habitability but also minimises the effects of the so-called "heat island". In our case, official native documentation has been considered and both species and their assembly through hydrozones have been proposed, which will greatly facilitate the estimation of requirements. If we limit these factors, we will reach a controlled demand for irrigation and reduce the negative effects of climate change, preventing negative impacts on the natural hydrology.

In conclusion, the present research study demonstrates, within a developed management model, that it is possible to reuse rainwater in a specific location, according to a methodology that develops a Project based on a Strategic Business Unit, recognising and analysing aforementioned variables, and aligning not only with the specific climate of the region of implementation but also with ongoing efforts to reach international goals.

Water is indispensable for the survival of living creatures and essential to ecosystems.

**Author Contributions:** Conceptualization, A.L.G., S.P.-M. and M.R.C.; methodology, A.L.G., S.P.-M. and M.R.C.; software, S.P.-M. and M.R.C.; validation, A.L.G., S.P.-M. and M.R.C.; formal analysis, A.L.G., S.P.-M. and M.R.C.; investigation, A.L.G., S.P.-M. and M.R.C.; resources, A.L.G., S.P.-M. and M.R.C.; writing—original draft preparation, A.L.G., S.P.-M. and M.R.C.; writing—review and editing, A.L.G. All authors have read and agreed to the published version of the manuscript.

**Funding:** This research received no external funding.

**Institutional Review Board Statement:** Not applicable.

**Informed Consent Statement:** Not applicable.

**Data Availability Statement:** Data are contained within the article.

**Conflicts of Interest:** Authors Sara Perales-Momparler and Miguel Rico Cortés were employed by the company Green Blue Management S. L. (TYPSA Group). The remaining authors declare that the research was conducted in the absence of any commercial or financial relationships that could be construed as a potential conflict of interest.

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
