# Peer review of "Proposal for Applying Sustainable Drainage Systems (SuDSs) as a Strategic Business Unit at a Military Development Located in Southern Europe (Córdoba, Spain): “Project BLET”"

_sustainability, doi:10.3390/su16052034_

Round 1

Reviewer 1 Report

Comments and Suggestions for Authors

The suggested project of water management sustainability seems interesting. Because, this is in the proposal level, it could be interesting to know, for example, to compare the mathematical calculations of water management not only with the InfoDrainage model, but e.g. SWMM system, or Mike DHI.

Author Response

Good evening.
I am sending you the document with the modifications in green color.
Thank you for your comments.

I'm sorry to say that we are not experts on other programs, but we have done a review (item 12)

Good afternoon. I send you a new version with actualized references. Thank you very much  (20240120_SustainableDrainageSystems_BLET-V02[17938])

Reviewer 2 Report

Comments and Suggestions for Authors

The paper ”Proposal for applying Sustainable Drainage Systems (SuDS), as a Strategic Business Unit, at a military development located in Southern Europe (Córdoba, Spain): “Project BLET”  brings to our attention a current topic regarding Sustainable Urban Drainage Systems (SuDS) and their implementation in a large-scale military development. In the context of the "Project BLET" in Córdoba, Spain, the proposal focuses on integrating SuDS as a Strategic Business Unit, aiming to address challenges associated with sustainable water management and provide an innovative model within military developments.

Project Context: "Project BLET" represents a significant transformation of 85 agricultural hectares into an industrial zone, where the Spanish Army intends to establish a new logistics base in Southern Europe. In light of Spain's commitments to the environment and sustainability on the international level, the proposal emphasizes the importance of SuDS in responsible infrastructure development.

While the paper extensively addresses the contemporary issue of Sustainable Urban Drainage Systems (SuDS) and their implementation within a large-scale military development, it requires restructuring and improvements for enhanced clarity.

In its current form, the paper is structured as a description/presentation/case study of a proposed project.

To enhance Sections 1-5, it is important to update and consolidate the bibliographic references and provide solid justifications for statements based on these references. Additionally, clarifying and highlighting the connections between the provided information and the respective sources is useful.

Point 7, entitled "Methodology and design criteria," primarily consists of a list of principles taken from R. Kellagher et al., "The SUDS manual," Department for Environment Food and Rural Affairs, Ciria, 2015. In this chapter, I recommend detailing your own research methods, design criteria, and numerical modeling approaches.

The paper should be structured according to research articles. The abstract and introduction should clearly define the research purpose and not just describe a context. Where and why will sustainable urban drainage systems (SUDS) be implemented, what is the aim, what were the brief research results, and what are the conclusions? I recommend a clear description of the research method, considering the conditions for implementing SUDS. Then, based on the research method, the results and conclusions should be clearly highlighted. I believe this type of research should specifically address requirements or conditions for the operation/viability of SUDS: where will they be located and why? What are the diffuse pollution sources within the complex? How will SUDS be sized to meet the needs? Are the pursued objectives justified in the specific climate of the implementation region? The paper specifies that water will be reused for irrigation and non-potable consumption. How will water be captured and stored with the SUDS regulator?

Aspects related to rainwater harvesting should be based on precise calculations: what is the impermeable surface from which rainwater will be collected in the specific analyzed case; what is the surface to be irrigated, what species it will contain, and what is the water demand, etc. All these data will lead to a accurate analysis of the economic and environmental benefits of Sustainable Urban Drainage Systems (SUDS).

In conclusion, you should highlight your own methods for designing, numerically modeling SUDS, and the calculations related to water requirements and collected quantities. This is essential to derive clear conclusions in line with the research methods and results. In its current form, the paper gives the impression of a concise review.

Author Response

Good evening.
I am sending you the document with the modifications in green color.
Thank you for your comments.

Good morning.

Thank you again for yor comments. I will explain the changes:

  • I have improved sections 1 to 5 with connections between them.
  • The article has been structured according to your indications (all of them in green colour).
  • The abstract and introduction have been expanded and reworded, objectives, a "literature review" has been included, referencing where surface and hydrozone data are obtained from, etc.
  • For the calculation of the impervious surface, I have followed legal documents published on the procurement platform and given an example of a logistics facility. 
    The methodology to be followed includes which plants to locate in hydrozones, according to local manuals. The hydrozones have been defined.

Although everything is referenced, the references format is not adequate. Due to my father's health problems, I am still working to adapt it.

Good afternoon. I send you a new version with actualized references. Thank you very much  (20240120_SustainableDrainageSystems_BLET-V02[17938])

Reviewer 3 Report

Comments and Suggestions for Authors

Dear Authors,

the work deals with an interesting topic. Research on rainwater management is obviously necessary. Sustainable exploitation of natural resources and their protection is of key importance for smart development. This study proposes a management model, wich includes water management as a strategic business unit, and it is intended to be a sustainable proposal which, from a strategic perspective, is able to consider rainfall as an alternative supply for non-potable water uses. The paper falls within the scope of Sustainability journal. In my opinion, such research is not novelty and has little interest in the international scale. There are plenty of scientific articles on this topic. But the outcomes can be beneficial for designing local strategies of rainwater management, especially for this case study (Project BLET).

The introduction provides an appropriate research background. The methodology was appropriately developed for the scope of the research. The main part of the article is description of case study and methodology. Authors have shown only a few results without their detailed analysis. There is no comparison to the results of other researchers. This should be added. The conclusions are described to general, it seems to be rather summary of the paper. 

The article also needs some corrections, for example:

- description of Figure 6,

- low resolution of Figure 6,

- there are many unnecessary spaces in the text, and some sentences are missing them.,

- all text should be formatted in accordance with the journal's guidelines.

Author Response

Good evening.
I am sending you the document with the modifications in green color.
Thank you for your comments.

Good morning.

Thank you very much for your comments again.

Good afternoon. I send you a new version with actualized references. Thank you very much  (20240120_SustainableDrainageSystems_BLET-V02[17938])

Although you have commented that the introduction and methodology are adequate, we have reformulated them.
Figure 6 has been described and its resolution has been increased.
Although everything is referenced, the format of the references is not adequate. Due to my father's health problems, I am working to adapt it.

Thank you very much

Reviewer 4 Report

Comments and Suggestions for Authors

Dear authors,

I have attached below my suggestions for your article.

All the best

Author Response

Good evening.
I am sending you the document with the modifications in green color.
Thank you for your comments.

Good morning,

  • The introduction and references in general have been expanded and reformulated. A review of the literature has been included.
    Although everything is referenced, the format of the references is not adequate. Due to my father's health problems, I am still working to adapt it.
  • An attempt has been made to express that the methodology is valid for any region.
  • A caption has been included in figure five.
  • I have reformulated results and conclusions

Thank you very much

Good afternoon. I send you a new version with actualized references. Thank you very much  (20240120_SustainableDrainageSystems_BLET-V02[17938])

Round 2

Reviewer 4 Report

Comments and Suggestions for Authors

Dear authors

I have seen the changes you have made. I hope that at this point you appreciate the increase in value of the article.

All the best

Author Response

Thank you very much for your comments and corrections. We appreciate the increase in value, of course.
